# Redox Homeostasis and Molecular Biomarkers in Precision Therapy for Cardiovascular Diseases

**DOI:** 10.3390/antiox13101163

**Published:** 2024-09-25

**Authors:** Cristina Manuela Drăgoi, Camelia Cristina Diaconu, Alina Crenguța Nicolae, Ion-Bogdan Dumitrescu

**Affiliations:** 1Faculty of Pharmacy, “Carol Davila” University of Medicine and Pharmacy, 020956 Bucharest, Romania; cristina.dragoi@umfcd.ro (C.M.D.); ion.dumitrescu@umfcd.ro (I.-B.D.); 2Faculty of Medicine, University of Medicine and Pharmacy Carol Davila Bucharest, 050474 Bucharest, Romania; camelia.diaconu@umfcd.ro; 3Department of Internal Medicine, Clinical Emergency Hospital of Bucharest, 105402 Bucharest, Romania

**Keywords:** oxidative stress, precision medicine, cardiovascular system, heart failure, mitochondrial dysfunction, redox homeostasis

## Abstract

Precision medicine is envisioned as the future of cardiovascular healthcare, offering a more tailored and effective method for managing cardiovascular diseases compared to the traditional one-size-fits-all approaches. The complex role of oxidative stress in chronic diseases within the framework of precision medicine was carefully explored, delving into the cellular redox status and its critical involvement in the pathophysiological complexity of cardiovascular diseases (CVDs). The review outlines the mechanisms of reactive oxygen species generation and the function of antioxidants in maintaining redox balance. It emphasizes the elevated reactive oxygen species concentrations observed in heart failure and their detrimental impact on cardiovascular health. Various sources of ROS within the cardiovascular system are examined, including mitochondrial dysfunction, which contributes to oxidative stress and mitochondrial DNA degradation. The article also addresses oxidative stress’s role in myocardial remodeling, a process pivotal to the progression of heart diseases. By integrating these aspects, the review underscores the importance of redox homeostasis and identifies molecular biomarkers that can enhance precision therapy for CVDs. The insights provided aim to pave the way for targeted therapeutic strategies that mitigate oxidative stress, thereby improving patient outcomes in cardiovascular medicine.

## 1. The Implications of Oxidative Stress in Chronic Diseases in the Era of Precision Medicine

Oxidative stress, characterized by an imbalance between the production of reactive oxygen species (ROS) and antioxidant defense systems, plays a critical role in the pathogenesis of metabolic and cardiovascular diseases. Precision medicine, which tailors medical treatment to the individual characteristics of each patient, has opened new paths for understanding and managing these diseases through the lens of oxidative stress [1,2,3].

In metabolic disorders such as diabetes mellitus, oxidative stress is a pivotal factor in the development and progression of insulin resistance and β-cell dysfunction. Elevated levels of ROS in diabetic patients lead to oxidative damage of cellular components, including lipids, proteins, and DNA, exacerbating insulin resistance and impairing insulin secretion. Precision medicine approaches aim to identify specific oxidative stress biomarkers and genetic predispositions that contribute to these oxidative mechanisms, enabling the development of targeted antioxidant therapies. For instance, genetic profiling can reveal variations in genes encoding antioxidant enzymes like superoxide dismutase (SOD) and glutathione peroxidase (GPx), guiding personalized interventions to enhance antioxidant capacity in diabetic patients [4,5,6,7,8].

In the nervous system, ROS are both signaling molecules and agents of damage. Physiologically, ROS participate in synaptic plasticity, neuronal differentiation, and adaptation to neuronal activity. However, an imbalance in ROS production and antioxidant defenses leads to oxidative stress, which has been implicated in neurodegenerative diseases such as Alzheimer’s, Parkinson’s, and amyotrophic lateral sclerosis (ALS). Excessive ROS can damage cellular components, including lipids, proteins, and DNA, triggering neuroinflammation and neuronal death. In the immune system, ROS are essential to the pathogen-killing mechanism of phagocytes, including neutrophils and macrophages. During the respiratory burst, these cells produce large amounts of ROS to destroy invading microorganisms. While beneficial in host defense, dysregulated ROS production can result in tissue damage and contribute to chronic inflammatory conditions such as rheumatoid arthritis, inflammatory bowel disease, and chronic obstructive pulmonary disease (COPD). ROS modulates the function of various immune cells, influencing signaling pathways that regulate cytokine production, cell proliferation, and apoptosis, thereby shaping immune responses in relation to a plethora of pathogenic cells, including oncogenic entities [9,10,11,12,13]. 

ROS also plays a significant role in the reproductive system. In sperm cells, controlled ROS levels are necessary for processes such as capacitation, hyperactivation, and the acrosome reaction, which are essential for successful fertilization. However, oxidative stress due to excessive ROS can lead to sperm dysfunction, DNA damage, and infertility. In the female reproductive system, ROS are involved in the regulation of folliculogenesis, ovulation, and luteal function. Imbalances in ROS levels are linked to conditions such as polycystic ovary syndrome (PCOS), endometriosis, and preeclampsia, which can affect fertility and pregnancy outcomes. Furthermore, in the skeletal system, ROS are involved in the regulation of bone remodeling. Osteoclasts generate ROS to degrade bone matrix during resorption, while osteoblasts require a delicate balance of ROS for differentiation and bone formation. Oxidative stress can disrupt this balance, leading to bone diseases such as osteoporosis. Elevated ROS levels enhance osteoclast activity and inhibit osteoblast function, contributing to bone loss and increased fracture risk [1,14]. 

In cardiovascular diseases, oxidative stress is implicated in atherosclerosis, hypertension, myocardial infarction, and heart failure. ROS contributes to endothelial dysfunction, a key event in atherogenesis, by reducing the bioavailability of nitric oxide (NO) and promoting inflammation. In the context of personalized medicine, identifying specific oxidative stress markers and genetic variants associated with ROS production and antioxidant defense can support pin point patients at higher risk for cardiovascular events. For example, polymorphisms in genes such as NADPH oxidase and endothelial NO synthase (eNOS) have been linked to increased oxidative stress and cardiovascular risk. Targeted therapies that address these genetic susceptibilities, such as NO donors or NADPH oxidase inhibitors, can be developed to mitigate oxidative damage and improve cardiovascular outcomes [15,16,17].

Moreover, precision medicine facilitates the identification of patient-specific environmental and lifestyle factors that contribute to oxidative stress. Factors such as diet, physical activity, and exposure to environmental noxious molecules can significantly influence oxidative stress levels. Personalized lifestyle modifications, informed by genetic and biomarker analysis, can effectively reduce oxidative stress and its detrimental effects on metabolic and cardiovascular health. For example, dietary recommendations rich in antioxidants or tailored exercise regimens can be prescribed based on an individual’s genetic profile and oxidative stress biomarkers [18,19,20,21,22].

A series of drugs are interfering with the general oxidative homeostasis. For example, statins, commonly known for their cholesterol-lowering properties by inhibiting HMG-CoA reductase, also demonstrate substantial antioxidant effects. Beyond lipid regulation, statins modulate oxidative stress by reducing ROS production and enhancing the body’s endogenous antioxidant defenses. One of the key mechanisms is through the upregulation of Nrf2 (Nuclear factor erythroid 2-related factor 2), a transcription factor responsible for activating the expression of antioxidant genes. Nrf2 activates the expression of heme oxygenase-1 (HO-1), an enzyme that plays a critical role in the antioxidant defense mechanism. Studies, such as the one by Mansouri et al. (2022), have shown that statins can modulate Nrf2/HO-1 signaling, reducing oxidative damage in various diseases, including cardiovascular conditions and neurodegenerative diseases.

Moreover, statins have been found to increase the activity of superoxide dismutase (SOD), an enzyme that converts superoxide radicals into less harmful molecules. The study by Daliri et al. (2023) systematically reviews the impact of statins on SOD levels, highlighting their role in reducing oxidative stress through enhanced superoxide neutralization. This antioxidant activity, combined with their anti-inflammatory properties, suggests that statins could be valuable not only in managing cardiovascular diseases but also in preventing oxidative damage in other organ systems.

In addition to statins, other pharmacological agents, such as ACE inhibitors, beta-blockers, and calcium channel blockers, exhibit antioxidant effects, often as a secondary benefit to their primary therapeutic actions. For example, ACE inhibitors reduce oxidative stress by decreasing the production of angiotensin II, a molecule that increases ROS production. By lowering angiotensin II levels, ACE inhibitors reduce endothelial dysfunction and protect tissues from oxidative damage.

Resveratrol, a natural polyphenol found in grapes, has gained attention for its potent antioxidant properties. It works by scavenging free radicals and enhancing endogenous antioxidant systems, such as glutathione peroxidase and catalase. In the context of cardiovascular health, Zivarpour et al. (2022) highlighted resveratrol’s potential in preventing cardiac fibrosis by mitigating oxidative damage and suppressing inflammatory pathways. Its combination with statins in clinical applications could be explored for synergistic effects in reducing oxidative stress.

The antioxidant potential of drugs, particularly statins, opens new avenues for treating conditions characterized by excessive oxidative stress, such as cardiovascular disease, neurodegenerative disorders, and certain cancers. Given their ability to modulate key signaling pathways involved in oxidative stress, statins, along with other antioxidants, may offer a multifaceted approach to disease management. Future research could focus on exploring drug combinations that enhance antioxidant defenses, targeting both ROS production and the body’s endogenous antioxidant mechanisms [23,24,25,26].

Advancements in omics technologies, including genomics, proteomics, and metabolomics, have been instrumental in elucidating the complex interplay between oxidative stress and metabolic and cardiovascular diseases. These technologies enable comprehensive profiling of oxidative stress-related pathways and the identification of novel biomarkers. Integrating multi-omics data with clinical information allows for the construction of predictive models and the development of precision therapies targeting oxidative stress [27,28].

All in all, oxidative stress is a fundamental contributor to the pathophysiology of metabolic and cardiovascular diseases. The era of precision medicine offers unprecedented opportunities to power our understanding of oxidative stress for the development of personalized preventive, diagnostic, and therapeutic strategies. By identifying patient-specific oxidative stress profiles and genetic susceptibilities, precision medicine can enhance the management of metabolic and cardiovascular diseases, ultimately improving patient outcomes and quality of life [29,30].

## 2. Cellular Redox Status in the Pathophysiological Complex of Cardiovascular Diseases

Heart failure is a multifaceted clinical syndrome stemming from structural or functional cardiac disorders that compromise the ventricle’s ability to fill or eject blood. The clinical manifestations of heart failure include fluid retention, resulting in pulmonary congestion and peripheral edema, and diminished cardiac output, which may limit physical capacity. This condition remains a leading cause of morbidity and mortality in industrialized nations and presents a growing public health concern, particularly due to an aging population and the rising prevalence of heart failure among older people. The primary etiologies of heart failure include myocardial infarction, hypertension, cardiomyopathy, and valvular heart disease. Post-myocardial infarction, the heart undergoes a pathophysiological adaptation known as cardiac remodeling, characterized by structural and functional changes, as well as alterations in the extracellular matrix of the non-infarcted myocardium. This remodeling process results in significant changes in heart morphology and volume, progressive ventricular dilatation, and deterioration of pump function. The mechanisms underlying the development and progression of heart failure are the subject of intense research, with a significant focus on the role of various signaling pathways, including the sympathetic nervous system and the renin-angiotensin-aldosterone system [31,32,33]. 

Extensive experimental and clinical research over the past two decades has identified oxidative stress as a pivotal factor in the progression to chronic heart failure, particularly following myocardial infarction. Cardiac remodeling, involving substantial changes in gene expression and protein function within both the extracellular matrix and cardiomyocytes, has been recognized as a fundamental process in this progression. Although initially adaptive, aiming to normalize stress and preserve contractile function, ventricular remodeling ultimately leads to dilatation, increased fibrosis, arrhythmias, and reduced ejection fraction. Studies have consistently shown that oxidative stress is elevated in heart failure, defined as an imbalance between the production of reactive oxygen species (ROS) and the antioxidant defense mechanisms. As a consequence of the excessive amount of ROS that resides at the cellular level, mainly superoxide radicals and hydrogen peroxide, there can emerge cell destructive events that can drive the lack of full functionality, DNA damage, lipidic peroxidation, protein impairment, and apoptosis. All these negative effects are at the basis of cardiovascular impairment origin, building up in time, and emerging at the beginning as minor, then major clinical symptoms, essentially affecting the contractile ability of the cardiac tissue due to the structural damage of the proteins implicated in the excitation-contractile interdependency. Nevertheless, the impact of high concentrations of reactive species exerts an activation signaling for hypertrophic kinases, transcription factors, and matrix metalloproteinases (MMPs), promoting extracellular matrix remodeling through upregulated apoptosis and cardiac fibroblast proliferation. The final outcome of adding continuous pieces to this pathological puzzle is the emergence of myocardial remodeling and heart failure diagnosis (Figure 1) [34,35,36,37]. 

Markers of oxidative stress are relevant in patients with chronic heart failure, correlating with myocardial dysfunction and the severity of the disease. The mechanisms by which oxidative stress impacts myocardial function include the degradation of cellular proteins and membranes, induction of cellular dysfunction, and promotion of cell death via apoptosis and necrosis. Emerging research in other organ systems suggests that ROS exert nuanced effects depending on their concentration, production site, and cellular redox status. For example, ROS can influence extracellular matrix remodeling through the activation of MMPs, which play a crucial role in maintaining the balance between matrix deposition and degradation, thereby affecting chamber size, shape, and function. Dysregulation of MMP activity or an imbalance between MMPs and their inhibitors is implicated in cardiac remodeling. 

Matrix metalloproteinases (MMPs) are a family of enzymes that play a critical role in tissue remodeling, inflammation, and extracellular matrix degradation, especially in cardiovascular diseases. The activation of MMPs by oxidants, such as reactive oxygen species (ROS), can occur through both direct and indirect mechanisms.

Direct activation of MMPs by oxidants involves the direct modification of the MMP zymogens (pro-MMPs). ROS can induce the oxidation of thiol groups in the pro-domain of MMPs, leading to the cleavage or conformational change that activates the enzyme. For example, hydroxyl radicals and hydrogen peroxide (H_2_O_2_) have been shown to directly oxidize cysteine residues in the pro-MMP2 and pro-MMP9 zymogens, converting them to their active forms. This activation increases MMP activity, contributing to tissue degradation and inflammation in cardiovascular diseases like atherosclerosis and heart failure.

Oxidants can also activate MMPs indirectly through the upregulation of inflammatory cytokines, growth factors, and signaling pathways that promote MMP expression and activation. For example, ROS can activate the NF-κB and AP-1 transcription factors, which in turn increase the transcription of MMP genes such as MMP-1, MMP-2, and MMP-9. Additionally, ROS can modulate signaling pathways involving mitogen-activated protein kinases (MAPKs) and tumor necrosis factor-alpha (TNF-α), both of which are known to enhance MMP activity indirectly by increasing inflammatory responses.

The indirect mechanisms are particularly significant in chronic cardiovascular conditions where prolonged oxidative stress promotes sustained inflammation and tissue remodeling [35]. 

The pharmacological inhibition of MMPs has shown promise in reducing secondary remodeling in chronic heart failure.

A minor quantity of superoxide radicals is typically produced as a byproduct during mitochondrial oxidative phosphorylation through the utilization of molecular oxygen. The superoxide anion (O_2_•^−^) is neutralized by either nitric oxide (NO) or superoxide dismutase (SOD). SOD enzymes facilitate the conversion of O_2_•^−^ into hydrogen peroxide (H_2_O_2_), which is then reduced to water by the action of glutathione peroxidase (GSHPx) and catalase, two enzymes that are vital for the cellular redox balance. In pathological conditions, H_2_O_2_ can be reduced by one electron to generate highly reactive hydroxyl radicals (HO•) via the Fenton reaction in the presence of iron or through the Haber–Weiss reaction involving the superoxide radical. Furthermore, the interaction between O_2_•^−^ and NO results in the reduction of protective NO molecule and the production of noxious peroxynitrite (ONOO^−^) radical [38].

Peroxiredoxins (PRDXs) are a family of thiol-specific peroxidases that play a critical role in maintaining cellular redox homeostasis by reducing peroxides, including hydrogen peroxide (H_2_O_2_) and organic hydroperoxides. These enzymes are highly conserved across species and are found in almost all cellular compartments, indicating their fundamental importance in cellular antioxidant defense. PRDXs act by utilizing a reactive cysteine residue at their active site to reduce peroxides, which in turn is regenerated through a thiol-based recycling system involving thioredoxin (TRX) and thioredoxin reductase (TRXR).

There are six known isoforms of peroxiredoxins (PRDX1–6), each with distinct cellular functions and localizations. PRDX1 and PRDX2 are ubiquitous in the cytoplasm, while PRDX3 is mitochondrial, and PRDX6 is notable for its ability to reduce phospholipid hydroperoxides, conferring protection to cellular membranes. PRDXs are not only efficient in detoxifying peroxides but also serve as sensors and regulators of hydrogen peroxide signaling, making them integral to redox signaling pathways that control cell proliferation, differentiation, and apoptosis.

Moreover, PRDXs have been linked to the regulation of inflammation and immune responses, where excessive peroxide accumulation can exacerbate inflammatory processes. Their multifunctional roles make PRDXs critical not only as antioxidant enzymes but also as modulators of oxidative stress signaling, particularly in pathological states such as cancer, neurodegenerative diseases, and cardiovascular diseases.

Catalase is one of the most efficient enzymes in nature, responsible for catalyzing the decomposition of hydrogen peroxide (H_2_O_2_) into water and oxygen. Found predominantly in the peroxisomes of eukaryotic cells, catalase plays a vital role in protecting cells from oxidative damage by maintaining low intracellular levels of H_2_O_2_, a byproduct of various metabolic reactions, particularly those involving the mitochondria and peroxisomes.

Catalase operates at a remarkably high turnover rate, capable of decomposing millions of hydrogen peroxide molecules per second, which makes it indispensable during periods of oxidative stress. In tissues with high metabolic activity, such as the liver and kidneys, catalase helps mitigate the effects of elevated ROS production. Unlike other antioxidant enzymes, catalase has a high Km for H_2_O_2_, meaning it is more active when H_2_O_2_ concentrations are elevated, such as during acute oxidative stress. This allows catalase to act as a safety net in conditions where ROS levels are rapidly increasing.

Catalase also works synergistically with other antioxidant systems. For example, superoxide dismutase (SOD) converts superoxide anions (O_2_•^−^) into hydrogen peroxide, which catalase subsequently breaks down. This collaboration between SOD and catalase helps prevent the accumulation of harmful ROS and limits oxidative damage to cellular components such as lipids, proteins, and DNA.

Both peroxiredoxins and catalase are crucial components of the cellular antioxidant defense network. While catalase is specialized in rapidly detoxifying hydrogen peroxide, peroxiredoxins offer versatility by reducing a broader range of peroxides and playing a regulatory role in redox signaling. Together, these enzymes help maintain redox homeostasis, protecting cells from oxidative stress and mitigating damage in various diseases characterized by excessive ROS production [39,40].

Studies have demonstrated a link between oxidative stress, matrix metalloproteinase activation, and left ventricular dilatation in patients with ischemic heart disease undergoing coronary artery bypass surgery. Researchers have reported a positive correlation between the left ventricular diastolic volume index and the activation of matrix metalloproteinases 2 and 9 in pericardial fluid during surgery. Furthermore, both metrics are positively correlated with levels of 8-isoprostaglandin F2α, a marker of oxidative stress. Although the number of studies is limited, these findings support the hypothesis that myocardial oxidative stress is a crucial marker of matrix metalloproteinase activation, contributing to ventricular remodeling and left ventricular dilatation in patients with ischemic heart disease [41,42,43].

Reactive oxygen species (ROS) significantly influence the pathophysiology of chronic heart diseases. The superoxide anion (O_2_•^−^) deactivates the protective nitric oxide (NO) molecule, diminishing its bioavailability and leading to severe endothelial dysfunction.

The non-enzymatic reaction of superoxide with certain molecules, such as nitric oxide (NO), is indeed faster than the enzymatic reaction catalyzed by superoxide dismutase (SOD). This reaction between superoxide and nitric oxide forms peroxynitrite (ONOO^−^), a highly reactive and damaging species. While SOD efficiently catalyzes the dismutation of superoxide into hydrogen peroxide and oxygen, the reaction between superoxide and nitric oxide is diffusion-limited, occurring almost instantaneously. This makes the non-enzymatic reaction significantly faster than the SOD-catalyzed pathway, which can have important implications in oxidative stress conditions, where peroxynitrite formation may lead to further cellular damage and inflammation.

This highlights the importance of regulating both enzymatic and non-enzymatic pathways to prevent oxidative damage in diseases such as cardiovascular disorders [44]. The reaction between NO and superoxide produces peroxynitrite, a strong oxidant. ROSs also engage in the process of redox signaling, impacting extracellular signaling pathways and molecules, which can lead to both immediate and long-term detrimental effects. Proteins essential for myocardial excitation-contraction coupling, such as ion channels, calcium release channels in the sarcoplasmic reticulum, and myofilaments, can be modified by redox reactions, altering their activity. Additionally, ROSs affect cellular energetics and play a vital role in the chronic changes in cell phenotype associated with heart failure. They regulate fibroblast proliferation, collagen synthesis, and the activation and expression of matrix metalloproteinases. Continuous exposure to ROSs mitigates secondary cardiac remodeling in experimental models of heart failure induced by coronary artery ligation. Redox-sensitive signaling pathways and transcription factors are involved in the development of cardiomyocyte hypertrophy, and antioxidant therapy has been demonstrated in clinical studies to slow the progression of cardiac hypertrophy and improve outcomes [45,46].

A critical area of investigation involves identifying the sources of ROS generation in the diseased heart and the factors regulating these processes, particularly concerning redox homeostasis. Potential sources of ROSs include inflammatory cell infiltration, mitochondria, xanthine oxidase, and NADPH oxidase. Mitochondrial ROS excess is likely to contribute significantly to contractile dysfunction in advanced stages of heart failure. An increase in xanthine oxidase expression and activity has been documented in end-stage human chronic heart failure. For instance, a study by Kameda et al. reported that patients with acute myocardial infarction treated with xanthine oxidase inhibitors, such as allopurinol, exhibited decreased plasma matrix metalloproteinase activity and reduced urinary levels of 8-isoprostaglandin F2α compared to those not treated with allopurinol. However, allopurinol may exert non-specific antioxidant effects and thus cannot be considered a specific indicator of xanthine oxidase activity in these studies [42,47].

Recent reports suggest that a significant source of ROS is the family of enzymes known as NADPH oxidases. Several pathophysiological stimuli involved in chronic heart failure, such as angiotensin II, α-adrenergic agonists, endothelin-1, and tumor necrosis factor, can stimulate ROS production by NADPH oxidase. Evidence of increased myocardial NADPH oxidase expression and activity has been demonstrated in patients with chronic heart failure. Studies using genetically altered models with defective NADPH oxidase activity have shown that NADPH oxidase plays a crucial role in angiotensin II signaling, contributing to cardiac hypertrophy and interstitial fibrosis. While these studies underscore the importance of NADPH oxidase in cardiac hypertrophy and heart failure, it is noteworthy that ROS from enzymatic sources can modulate or trigger the activity of other ROS sources [48,49,50].

NADPH oxidases (NOX) are a family of enzymes that play a pivotal role in producing reactive oxygen species (ROS) as signaling molecules. Angiotensin II, a key regulator in cardiovascular physiology, particularly in hypertension and heart failure, stimulates NOX isoforms to generate ROS, which in turn contributes to several downstream pathophysiological effects such as oxidative stress, inflammation, and fibrosis. Specific NOX isoforms are involved in distinct processes within angiotensin II signaling.

NOX1 is highly expressed in vascular smooth muscle cells and is activated by angiotensin II. It plays a central role in the vasoconstrictive and pro-inflammatory effects of angiotensin II. Upon angiotensin II stimulation, NOX1 generates ROS, leading to the activation of mitogen-activated protein kinases (MAPKs) and the upregulation of inflammatory cytokines, which contribute to vascular remodeling, hypertension, and atherosclerosis.

NOX2, predominantly found in endothelial cells and immune cells, is also activated by angiotensin II. In the context of cardiovascular disease, NOX2-mediated ROS production plays a role in endothelial dysfunction, a hallmark of hypertension and atherosclerosis. NOX2-derived ROS contribute to the activation of pro-inflammatory pathways, endothelial nitric oxide synthase (eNOS) uncoupling, and the promotion of oxidative stress, further enhancing angiotensin II’s deleterious effects on the vasculature.

NOX4 is expressed in many cardiovascular tissues, including vascular smooth muscle cells, endothelial cells, and cardiomyocytes. Unlike NOX1 and NOX2, which produce superoxide, NOX4 primarily produces hydrogen peroxide (H_2_O_2_). In the setting of angiotensin II signaling, NOX4 plays a dual role, acting both as a mediator of oxidative stress and as a regulator of adaptive responses. In the heart, NOX4 is involved in angiotensin II-induced cardiac hypertrophy and fibrosis, and its activity contributes to the activation of fibrotic signaling pathways such as transforming growth factor-beta (TGF-β).

NOX1 and NOX2 promote vascular inflammation and remodeling, exacerbating conditions such as hypertension and atherosclerosis. NOX4, while promoting fibrosis, also contributes to adaptive responses in some contexts, highlighting its complex role in vascular health.

NOX4 is particularly implicated in angiotensin II-induced cardiac hypertrophy and fibrosis, while NOX2 can contribute to oxidative stress and myocardial dysfunction.

Overall, these NOX isoforms are key mediators of oxidative stress and inflammatory responses driven by angiotensin II in cardiovascular diseases. Understanding their specific roles opens new avenues for therapeutic targeting in conditions like hypertension and heart failure [51].

## 3. Generation of Reactive Oxygen Species and Antioxidants

The equilibrium between the production of reactive oxygen species (ROSs) and their elimination by antioxidant systems constitutes the homeostasis of the redox processes. Oxidative stress is defined as the excessive production of reactive oxygen species relative to antioxidant levels. Reactive oxygen species are oxygen-based chemical entities with heightened reactivity, including free radicals such as superoxide (O_2_•^−^) and hydroxyl (HO•) radicals, as well as non-radical species capable of generating free radicals, such as hydrogen peroxide (H_2_O_2_). Superoxide is a primary radical that can lead to the formation of other reactive oxygen species, such as hydrogen peroxide and hydroxyl radicals. The hydroxyl radical is also generated by the reduction of hydrogen peroxide in the presence of endogenous iron via the Fenton reaction. 

Nitric oxide is essential for normal cardiac physiology, regulating cardiac functions such as coronary vasodilation, platelet inhibition, neutrophil adhesion and activation, and modulation of contractile cardiac function. NO also plays a protective role against ischemia and/or heart failure through various mechanisms, including the stimulation of soluble guanylate cyclase, which decreases intracellular calcium concentrations and inhibits oxidative stress. Therefore, the superoxide radical can exert direct cytotoxic effects by inactivating cytoprotective NO and forming highly reactive species such as peroxynitrite (ONOO-) as a result of the interaction between NO and superoxide [52,53,54,55,56].

Several specific and non-specific antioxidant defense systems act in order to degrade reactive oxygen species into non-toxic molecules. Under physiological conditions, these toxic effects can be mitigated by enzymes such as superoxide dismutase (SOD), glutathione peroxidase (GSHPx), and catalase, as well as non-enzymatic antioxidants (Figure 2).

Glutathione peroxidase 1 (GPX1) is a selenium-containing enzyme that plays a critical role in protecting cells from oxidative damage by reducing hydrogen peroxide (H_2_O_2_) and organic hydroperoxides to water and alcohol, respectively. GPX1 uses reduced glutathione (GSH) as an electron donor in a process that operates via a ping-pong (bi-bi) mechanism.

In a ping-pong mechanism, the enzyme alternates between two states: after reacting with the first substrate (H_2_O_2_), it forms a covalent intermediate (in this case, the selenol of GPX1 is oxidized to selenenic acid). The enzyme is then restored to its active form through the reaction with the second substrate (GSH), which reduces the selenenic acid back to the selenol form, regenerating GPX1 for the next catalytic cycle.

The reaction between GPX1 and hydrogen peroxide is diffusion-limited, meaning that the rate at which GPX1 reduces hydrogen peroxide is primarily determined by how quickly hydrogen peroxide can diffuse to the enzyme’s active site. This high catalytic efficiency ensures that GPX1 can rapidly neutralize low levels of H_2_O_2_ before they cause significant oxidative damage. However, under high oxidative stress conditions, the enzyme may become saturated with hydrogen peroxide, affecting the overall rate of detoxification.

The Km (Michaelis constant) of GPX1 for GSH is an important factor in determining the enzyme’s catalytic efficiency. Km represents the concentration of GSH at which the reaction rate is half of its maximum velocity. GPX1′s Km for GSH is not fixed and depends on the intracellular concentration of GSH, which can vary under different physiological and pathological conditions.

In healthy cells with adequate levels of GSH, GPX1 operates efficiently, maintaining redox homeostasis by keeping H_2_O_2_ levels low. However, in conditions of oxidative stress, where GSH is depleted, the Km for GSH increases, indicating reduced enzyme efficiency. Since GSH is required to regenerate GPX1 in the catalytic cycle, low GSH levels can impair the ability of GPX1 to detoxify H_2_O_2_, leading to an accumulation of ROS and further oxidative damage [57]. GSHPx is a crucial antioxidant that catalyzes the reduction of hydrogen peroxide to water and lipid hydroperoxides to their corresponding alcohols, preventing the formation of other toxic radicals, such as the hydroxyl radical. GSHPx has a higher affinity for hydrogen peroxide than catalase and is abundantly present in the heart, particularly in the cytosolic and mitochondrial compartments, underscoring its importance as a defense mechanism. Consequently, GSHPx is expected to provide greater protection against oxidative damage compared to SOD. Studies have shown that mice overexpressing the GSHPx gene exhibit increased resistance to myocardial oxidative stress, as well as to remodeling and heart failure [58,59,60,61,62].

Melatonin, a multifaceted hormone produced primarily by the pineal gland, plays both direct and indirect roles as an antioxidant, providing a broad spectrum of protective effects against oxidative stress. Its direct antioxidant action involves the neutralization of reactive oxygen species (ROS) and reactive nitrogen species (RNS), thereby preventing cellular damage. Melatonin’s molecular structure allows it to interact directly with these free radicals, transforming them into less reactive molecules. This direct scavenging capacity extends to various ROS and RNS, including hydroxyl radicals, hydrogen peroxide, singlet oxygen, and peroxynitrite. This wide-ranging scavenging activity helps protect cellular components such as lipids, proteins, and DNA from oxidative damage, which can lead to cell death or mutations, and its role is highly evident in a broad spectrum of diseases and in most diverse tissues [63,64,65,66,67,68,69,70,71].

**Figure 2 antioxidants-13-01163-f002:**
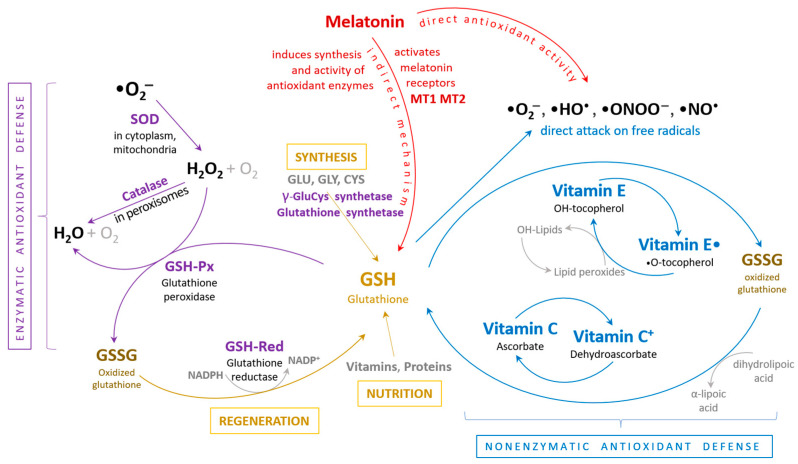
The body’s antioxidant defense mechanism entails a synergistic interaction between enzymatic and non-enzymatic antioxidant systems to collectively protect cells and organ systems from damage caused by free radicals. ROS (reactive oxygen species), SOD (superoxide dismutase), GSH (reduced glutathione), GSSG (oxidized glutathione), GSH-Px (glutathione peroxidase), GSH-Red (glutathione reductase) [72].

In addition to its direct antioxidant effects, melatonin enhances the body’s endogenous antioxidant defense mechanisms, constituting its indirect role. It stimulates the synthesis and activity of key antioxidant enzymes like superoxide dismutase (SOD), catalase, and glutathione peroxidase. These enzymes play crucial roles in neutralizing superoxide anions, hydrogen peroxide, and lipid peroxides, respectively, thereby reducing oxidative stress. Melatonin also upregulates the production of glutathione, a major intracellular antioxidant, by boosting the expression of gamma-glutamylcysteine synthetase, the rate-limiting enzyme in glutathione synthesis. This increase in glutathione levels enhances the cell’s capacity to detoxify reactive intermediates and maintain redox balance [72,73].

Melatonin’s indirect antioxidant role extends further through its interaction with the mitochondrial function. Mitochondria are significant sources of ROS during cellular respiration, and excessive ROS production can lead to mitochondrial dysfunction and apoptosis. Melatonin stabilizes mitochondrial membranes, reduces electron leakage, and thus diminishes ROS generation. Moreover, melatonin supports mitochondrial biogenesis and function by activating signaling pathways such as the SIRT1-PGC-1α axis, which enhances the production of new mitochondria and improves their efficiency. By maintaining mitochondrial health, melatonin ensures that cells can produce energy efficiently without generating excessive ROS, thus protecting against oxidative damage indirectly. Through these comprehensive actions, melatonin serves as a potent and versatile antioxidant, safeguarding cellular integrity and function across various biological systems [65,74,75].

At the cardiovascular level, melatonin’s antioxidant effects are especially beneficial. The cardiovascular system is highly susceptible to oxidative stress due to its constant exposure to high levels of oxygen and the sheer volume of blood it processes. Oxidative stress is a key contributor to endothelial dysfunction, a precursor to atherosclerosis and other cardiovascular diseases. Melatonin mitigates this by enhancing the expression and activity of endogenous antioxidant enzymes such as superoxide dismutase (SOD), catalase, and glutathione peroxidase. Additionally, melatonin reduces the oxidation of low-density lipoproteins (LDL), which are pivotal in the formation of atherosclerotic plaques. Through these mechanisms, melatonin helps maintain endothelial health and prevents the progression of cardiovascular diseases. Furthermore, melatonin exerts protective effects against ischemia-reperfusion injury, a condition characterized by oxidative stress following the restoration of blood supply to previously ischemic tissues. In the cardiovascular context, ischemia-reperfusion injury is commonly associated with myocardial infarction. Melatonin administration has been shown to reduce infarct size and improve cardiac function post-infarction by diminishing oxidative stress and inflammation. It modulates the expression of pro-inflammatory cytokines and reduces the infiltration of neutrophils and macrophages, which are sources of further oxidative damage. Thus, melatonin not only acts as a potent antioxidant but also modulates inflammatory responses, offering comprehensive protection to the cardiovascular system against oxidative stress-related damage [76,77,78,79].

When the production of reactive oxygen species surpasses the antioxidant defense capacity, oxidative stress exerts detrimental effects on the structural and functional integrity of biological tissues. In the heart, specifically, excessive reactive oxygen species can induce myocardial remodeling, resulting in contractile dysfunction and structural alterations. Oxidative stress has also been implicated as a major mechanism of endothelial dysfunction, not only in atherosclerosis but also in heart failure. Clinical studies have demonstrated that endothelial dysfunction is independently associated with long-term adverse outcomes in patients with heart failure [50,80].

## 4. Elevated Reactive Oxygen Species Concentration in Heart Failure

Clinical and experimental studies have consistently demonstrated an increase in the generation of reactive oxygen species (ROS) in heart failure. Most experimental research involving various animal models of heart failure has been conducted on young animals without comorbid risk factors such as hypertension. However, evidence strongly supports that oxidative stress is elevated in heart failure and plays a critical role in its development and progression.

In the study of heart failure, animal models serve to replicate the phenotypes observed in clinical heart failure in human patients. Two primary types of animal models were utilized: rapid pacing-induced heart failure in dogs and post-myocardial infarction heart failure in mice. These models exhibited structural and functional/hemodynamic characteristics similar to those seen in human heart failure patients. Research has revealed a negative correlation between malondialdehyde levels and left ventricular ejection fraction, as well as increased levels of lipoperoxide and 8-isoprostaglandin-F2α, which are biomarkers of ROS generation, in the plasma and pericardial lysate of heart failure patients [80,81,82,83].

Oxidative stress arises from an imbalance between ROS generation and the body’s antioxidant defense mechanisms. Thus, impaired antioxidant defenses, including enzymes like superoxide dismutase (SOD), catalase, and glutathione peroxidase (GSHPx), or reduced concentrations of endogenous antioxidants such as vitamin E, ascorbic acid, and glutathione, can elevate ROS levels. Studies have shown that heart failure secondary to myocardial infarction is associated with both antioxidant deficiency and increased oxidative stress.

Antioxidant deficiency refers to an insufficient capacity to neutralize reactive oxygen species (ROS) and other free radicals, leading to oxidative stress and potential damage to cells, proteins, lipids, and DNA. This deficiency can be chemical, enzymatic, or a combination of both, depending on the underlying mechanisms.

This type of deficiency occurs when there is an inadequate level of non-enzymatic antioxidants, which are small molecules that directly scavenge free radicals. These chemical antioxidants include vitamins (e.g., vitamin C, vitamin E), glutathione (GSH), and other dietary antioxidants like flavonoids and carotenoids. Deficiency can arise due to poor dietary intake, malabsorption, or increased demand for these antioxidants in conditions of oxidative stress (e.g., during chronic inflammation or disease). Without sufficient chemical antioxidants, the body lacks the ability to neutralize free radicals directly, leading to oxidative damage.

Enzymatic antioxidant deficiency refers to the impaired function or low activity of antioxidant enzymes that catalyze the conversion of ROS into less harmful molecules. Key enzymatic antioxidants include superoxide dismutase (SOD), catalase, glutathione peroxidase (GPX), and thioredoxin (TRX). These enzymes play a critical role in detoxifying superoxide radicals, hydrogen peroxide, and other reactive species. Deficiency may occur due to genetic mutations (e.g., in the SOD1 gene), reduced expression of these enzymes, or insufficient co-factors necessary for their activity (e.g., selenium for GPX). Enzymatic deficiencies compromise the body’s ability to enzymatically detoxify ROS, leading to a buildup of oxidative stress.

In many pathological conditions, both chemical and enzymatic antioxidant defenses may be impaired. For example, in chronic diseases such as cardiovascular disorders or neurodegenerative diseases, the overproduction of ROS may overwhelm both the chemical antioxidants and the enzymatic defense systems, leading to a compounded state of antioxidant deficiency. This combination exacerbates oxidative damage and accelerates disease progression.

Antioxidant deficiency can arise from a lack of chemical antioxidants, impaired enzymatic antioxidant function, or a combination of both. Addressing this deficiency requires a comprehensive approach, including dietary supplementation with chemical antioxidants and strategies to restore or enhance enzymatic antioxidant activity [84,85].

These changes have been correlated with hemodynamic function, suggesting their role in pathogenic cardiac dysfunction. Interestingly, there were no observed decreases in enzyme activity, including that of SOD and catalase, indicating that oxidative stress in heart failure may primarily result from increased ROS generation. These findings collectively highlight the critical role of oxidative stress in the pathophysiology of heart failure, emphasizing the need for further research into antioxidant-based therapeutic strategies to mitigate the adverse effects of elevated ROS levels [86,87].

## 5. Sources of Reactive Oxygen Species in the Cardiovascular System

Cellular sources of reactive oxygen species in the heart include cardiomyocytes, endothelial cells, and neutrophils. In cardiomyocytes, ROS can be produced by mitochondria, NAD(P)H oxidase, xanthine oxidase, and nitric oxide synthase (NOS). Mitochondria generate ROS by transferring single electrons to molecular oxygen during the respiratory chain process (Figure 3). Under normal physiological conditions, small amounts of ROS formed during mitochondrial respiration are detoxified by endogenous scavenging mechanisms. Studies have shown that inhibiting electron transport at complexes I and III increases superoxide radical production. In heart failure, mitochondria produce more O_2_•^−^ in the presence of NADH compared to normal mitochondria, which is associated with decreased enzyme complex activity. This highlights the pathophysiological link between mitochondrial dysfunction and oxidative stress. During ischemia or hypoxia, mitochondrial ROS production increases, contributing to myocyte destruction and myocardial infarction. Mitochondrial electron transport relies on a series of cytochrome-dependent enzymes located in the inner mitochondrial membrane, including NADH dehydrogenase (complex I), cytochrome b-c1 oxidase (complex III), cytochrome oxidase (complex IV), and small molecules of coenzyme Q. Electrons are transferred from NADH^+^ to oxygen via these complexes, and more than 98% of these electrons are used in ATP production, while 1–2% escape to form O_2_•^−^, which is removed by SOD. Blocking electron transport at complexes I or III leads to a substantial increase in peroxide radical formation [88,89,90].

In vascular endothelial cells and activated leukocytes, ROSs can be generated by NAD(P)H oxidase or xanthine oxidase. Each member of the NAD(P)H oxidase family contains a Nox catalytic unit that forms a heterodimer with a low molecular mass subunit called p22phox, facilitating electron transfer from NAD(P)H to O_2_, resulting in O_2_•^−^ formation. Nox1 and Nox2 require cytosolic subunits for activation, while Nox4 does not. Nox1 is prevalent in vascular smooth muscle cells, while Nox2 and Nox4 are found in cardiac myocytes, endothelial cells, and fibroblasts. Pathophysiologically relevant stimuli in heart failure, such as mechanical stretch, angiotensin II, endothelin-1, and tumor necrosis factor-α, significantly increase NAD(P)H oxidase activity and O_2_•^−^ production. Experimental models have shown that NAD(P)H oxidase deficiency, such as in p47phox-deficient animals, protects against left ventricular remodeling and dysfunction after myocardial infarction. Angiotensin II can exacerbate mitochondrial dysfunction by activating NAD(P)H oxidases in vascular endothelial cells, increasing mitochondrial ROS production and decreasing endothelial NO• availability. Nox2 and Nox4 are the primary isoforms implicated in diseased myocardium, with recent studies indicating Nox4’s significant role in mitochondrial ROS production and cardiac remodeling due to increased pressure and aging. The role of Nox5 in heart failure remains unclear [91,92,93,94].

Xanthine oxidase activity and quantity are elevated in heart failure, and treatment with xanthine oxidase inhibitors, such as allopurinol, has been shown to improve left ventricular function and myocardial efficiency. Chronic allopurinol treatment significantly reduces left ventricular remodeling post-myocardial infarction. These effects may involve the reduction of myocardial oxygen consumption and improvement in cardiac efficiency through NO inactivation. Unregulated NOS can lead to additional ROS production through the oxidation of the essential co-factor tetrahydrobiopterin (BH4). Endothelial NOS (NOS3) plays a critical role in pathological cardiovascular remodeling, including heart failure. In normal conditions, Nitric oxide synthase 3 (NOS3) synthesizes nitric oxide (NO) and L-citrulline from L-arginine and oxygen, but under oxidative stress or when BH4 or L-arginine levels are low, NOS3 becomes destabilized and produces even more reactive oxygen species, instead. This uncoupled NOS3 in endothelial cells and cardiac myocytes contributes to left ventricular remodeling under chronic stress. Animal studies show that reduced BH4 levels and uncoupled NOS3 are linked to the high incidence of left ventricular dilatation and dysfunction, which can be partially mitigated with BH4 supplementation [37,95,96,97].

Cytochrome c oxidase (COX), the last enzyme in the mitochondrial electron transport chain, is made up of 13 subunits. The subunits I, II, and III, encoded by mitochondrial DNA, are essential for the complex’s proper assembly and function. Myocardial infarction leads to decreased activity of enzymes in complexes I, III, and IV. Overexpression of COX III results in reduced COX I and COX activity and increased apoptosis, followed by heart failure. Leukocyte contribution to ROS generation is suggested by the presence of myeloperoxidase (MPO) in plasma, correlating with heart failure severity. Plasma MPO indicates increased leukocyte activation rather than systemic inflammation [98,99,100,101].

Another model for oxidative stress in CVD is organoids, which are three-dimensional (3D) structures derived from stem cells that replicate key aspects of their tissue of origin and have become invaluable tools in biomedical research. In the context of pharmacology, organoids are particularly useful for studying disease mechanisms, including oxidative stress, and for advancing drug discovery efforts, especially in complex conditions such as cardiovascular diseases (CVD).

Oxidative stress plays a central role in the pathophysiology of cardiovascular diseases by contributing to processes such as endothelial dysfunction, inflammation, and tissue damage. Reactive oxygen species (ROS) and reactive nitrogen species (RNS) generated during oxidative stress lead to the modification of lipids, proteins, and DNA, promoting atherosclerosis, heart failure, and hypertension. Understanding how oxidative stress affects cardiovascular tissues is key to identifying therapeutic targets.

Organoids derived from cardiovascular tissues—such as cardiac organoids or vascular organoids—provide a platform to study oxidative stress in a more physiologically relevant setting compared to traditional two-dimensional (2D) cell cultures. They mimic the 3D architecture and cellular diversity of the heart and blood vessels, allowing researchers to observe how oxidative stress impacts different cell types, including cardiomyocytes, endothelial cells, and smooth muscle cells. This is crucial for understanding how oxidative damage contributes to the onset and progression of cardiovascular diseases at the tissue level.

Cardiovascular organoids provide a microenvironment that closely resembles human tissue, allowing for more accurate modeling of oxidative stress mechanisms. They capture interactions between different cell types, which are essential for understanding how ROS/RNS affect tissue function. This is particularly important for studying the complex intercellular signaling involved in cardiovascular oxidative damage.

Organoids can be generated from patient-derived stem cells, allowing for personalized models of oxidative stress. This is especially beneficial for studying genetic predispositions to oxidative stress in cardiovascular diseases. For example, organoids from patients with specific genetic mutations associated with oxidative stress (e.g., mutations in antioxidant enzymes like SOD or catalase) can help in identifying patient-specific vulnerabilities and testing tailored antioxidant therapies.

Organoids allow for real-time monitoring of oxidative stress and redox status. Using advanced imaging techniques and redox-sensitive fluorescent probes; researchers can track ROS levels and antioxidant responses within the 3D organoid structure. This makes it easier to study the effects of oxidative stress on cellular and tissue function over time, providing insights into both acute and chronic oxidative damage in cardiovascular disease progression.

Organoids offer significant advantages in the field of drug discovery, particularly for identifying novel antioxidant therapies for cardiovascular diseases. Their ability to replicate the complexity of human tissues makes them ideal for testing how small molecules, gene therapies, or biologics affect oxidative stress and redox homeostasis in a realistic, human-relevant context.

Cardiovascular organoids can be used for high-throughput screening of potential antioxidant drugs. By exposing organoids to various compounds, researchers can observe their effects on oxidative stress markers, ROS/RNS levels, and cellular function. This accelerates the identification of lead compounds that may reduce oxidative damage in cardiovascular tissues.

Organoids provide a more predictive model for evaluating the efficacy and toxicity of drugs compared to traditional cell cultures or animal models. Because organoids capture the complex cellular interactions of human cardiovascular tissues, they are better suited for identifying potential off-target effects or toxicity of new antioxidant drugs. This is particularly important for cardiovascular therapies, where drug-induced cardiotoxicity is a major concern.

In addition to small molecule antioxidants, organoids can be used to test innovative therapeutic strategies such as gene therapies aimed at enhancing endogenous antioxidant defenses (e.g., upregulating Nrf2 or SOD pathways) or cell-based therapies involving stem cells engineered to resist oxidative stress. Organoids offer a controlled environment to assess how these therapies modulate oxidative stress and improve tissue function.

Organoids can help unravel the mechanisms by which oxidative stress contributes to specific cardiovascular conditions. For instance, they can be used to model how oxidative stress influences processes like fibrosis, angiogenesis, or inflammation in heart failure or atherosclerosis. Understanding these mechanisms is critical for developing therapies that target the root causes of oxidative damage in cardiovascular diseases.

The use of organoids in oxidative stress research and drug discovery is still evolving. Future advancements, such as the integration of vascularization in organoids and the development of bioengineered organoid platforms that mimic the biomechanical forces of the cardiovascular system, will further improve their utility. Additionally, combining organoids with other cutting-edge technologies like CRISPR for gene editing or single-cell RNA sequencing can help identify new therapeutic targets related to oxidative stress [102,103,104].

## 6. Oxidative Stress and Mitochondrial DNA Degradation

Reactive oxygen species (ROS) can inflict damage on mitochondrial macromolecules at or near their sites of formation. Mitochondria possess their own genomic system, known as mitochondrial double-stranded DNA (mtDNA), which consists of two strands: the light chain and the heavy chain. Transcription of mtDNA results in the production of messenger RNA, which encodes 13 essential subunits involved in oxidative phosphorylation. Additionally, transcription of the light chain generates primary RNA necessary for the initiation of mtDNA replication. The regulation of mitochondrial function is orchestrated by mtDNA and factors that control its transcription and replication, such as mitochondrial transcription factor A. Mitochondrial DNA is particularly susceptible to ROS-mediated degradation. This vulnerability arises from three primary factors: firstly, mitochondria lack the protective chromatin structure found in nuclear DNA; secondly, mtDNA exhibits limited repair capacity; and thirdly, superoxide radicals generated within mitochondria are unable to diffuse across membranes, leading to localized degradation. Consequently, mtDNA accumulates higher levels of 8-hydroxydeoxyguanosine, an oxidative modification, compared to nuclear DNA. Unlike nuclear-encoded genes, the expression of mitochondrial-encoded genes is regulated by the mtDNA copies number. Mitochondrial damage is manifested through degraded DNA, reduced mRNA transcription, impaired protein synthesis, and compromised mitochondrial function [105,106].

In heart failure, increased ROS production is linked to mitochondrial degradation. This degradation is characterized by elevated levels of lipid peroxidation within mitochondria, a reduction in mtDNA copies number, diminished mRNA transcription, and decreased oxidative capacity due to impaired enzyme complex activity. It is hypothesized that low levels of mtDNA, mRNA, and mitochondrial proteins are correlated with increased ROS production. Current research aims to quantify ROS using electron spin resonance (ESR) techniques and to investigate changes in mtDNA copy number and the activity of electron transport chain complexes [107,108].

## 7. Oxidative Stress in the Process of Myocardial Remodeling

The production of relatively low levels of reactive oxygen species plays a crucial role in modulating the activity of various intracellular molecules and signaling pathways, known as redox signaling, which can induce specific cellular phenotypic changes. Oxidative stress directly affects cell function and structure, integrally activating molecular signals involved in myocardial remodeling and failure. It stimulates myocardial growth, matrix remodeling, and cellular dysfunction through the activation of various signaling pathways.

Firstly, ROSs activate a wide array of transcription factors, hypertrophic signals, and kinases. Reactive species stimulate tyrosine kinase, protein kinase C, and the mitogenic activation of protein kinases. Low levels of hydrogen peroxide (H_2_O_2_) are associated with mitogenic activation of protein kinase and protein synthesis, whereas high levels activate c-Jun N-terminal kinase (JNK) and protein kinase B (Akt), leading to apoptosis.

Secondly, ROSs induce apoptosis, contributing to remodeling and dysfunction. This process is mediated by ROS-induced DNA and mitochondrial degradation and the activation of proapoptotic signaling kinases. ROSs cause DNA breakage and activate nuclear enzymes such as poly (ADP-ribose) polymerase-1 (PARP-1), which regulates the expression of various inflammatory mediators that facilitate cardiac remodeling progression.

Thirdly, ROSs activate matrix metalloproteinases (MMPs), a family of proteolytic enzymes. MMPs are generally secreted in an inactive form and are post-translationally activated by ROS. Additionally, ROSs stimulate the transcription of enzymes to enhance MMP expression. MMPs are pivotal in normal tissue remodeling processes, including cell migration, invasion, proliferation, and apoptosis. Increased MMP activation has been observed in heart failure, and MMP inhibitors have been shown to limit left ventricular dilatation following experimental myocardial infarction. Consequently, a proposed mechanism for left ventricular remodeling involves MMP activation secondary to increased ROS. Sustained MMP activation can alter myocardial structural properties by creating an abnormal extracellular environment for myocytes [109,110,111,112,113].

Furthermore, hydroxyl radicals and dimethyl thiourea inhibit MMP-2 activation, which is associated with remodeling and failure post-myocardial infarction. These findings suggest that elevated oxidative stress may stimulate myocardial MMP activation, playing a critical role in heart failure development and progression. Studies have shown that ROSs influence contractile function by altering proteins involved in excitation–contraction coupling [114,115,116].

## 8. Conclusions and Future Perspectives

Precision medicine represents the future of healthcare, particularly in the context of cardiovascular diseases, offering the potential for more efficient management due to the gradual onset, heterogeneous nature, and multimorbidity of these conditions. The pathogenesis of cardiovascular diseases can begin decades before clinical manifestations, making the use of precisely targeted diagnostic and therapeutic tools revolutionary for prevention, early diagnosis, and personalized treatment. Oxidative stress has a critical role in the development and progression of metabolic and cardiovascular diseases, making it a key target in precision medicine. Addressing oxidative stress through personalized interventions can significantly enhance treatment outcomes. Although precision medicine is still an evolving field with many technologies in their early stages of development, its promise is significant. Organoids represent a promising tool for advancing our understanding of oxidative stress in cardiovascular diseases and for developing novel, precise antioxidant therapies. By providing a more accurate and personalized model of human tissues, they are likely to play a key role in the future of precision medicine for cardiovascular health.

Current research and data on precision medicine face limitations due to ethical, social, legal, and economic challenges, but as the capacity and infrastructure to utilize new tools in cardiovascular disease management grow, the prospect of delivering tailored therapies to meet the unique needs and limitations of individual patients will transition from a hopeful vision to a fundamental responsibility.

The future of personalized cardiology approaches is imminent; it necessitates assimilation, adaptation, and enhanced accessibility through profound education and infrastructure development.

## Figures and Tables

**Figure 1 antioxidants-13-01163-f001:**
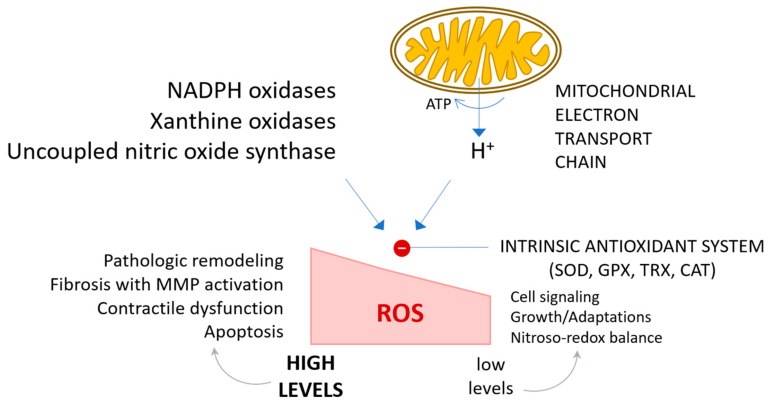
General overview of reactive oxygen species (ROS) generation pathways and antioxidant systems in the cardiac tissue. ROS generation and the corresponding antioxidant defense mechanisms in the heart are schematically illustrated. At physiological levels, ROS are believed to be vital for physiological cardiac signaling, adaptive growth responses, and extracellular matrix remodeling. However, elevated ROS levels are implicated in pathological cardiac remodeling, apoptosis, and chamber dysfunction. Key antioxidant systems include superoxide dismutase (SOD), glutathione peroxidase (GPX), thioredoxin (TRX), and catalase (CAT) [37].

**Figure 3 antioxidants-13-01163-f003:**
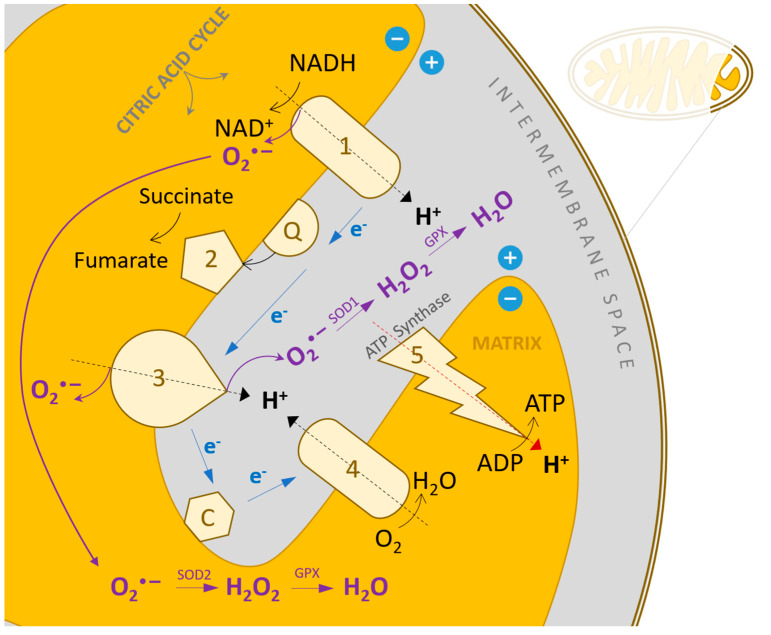
Mitochondrial electron transport chain, a major physiological source of pro-oxidant species and the compensatory mechanisms of neutralization [37].

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
