# Peer review of "Redox Homeostasis and Molecular Biomarkers in Precision Therapy for Cardiovascular Diseases"

_antioxidants, 2024, doi:10.3390/antiox13101163_

Round 1

Reviewer 1 Report

This is a well written review on this topic. However, I have some comments/suggestions to the authors. One is to reduce the number of self-citations. I would also suggest to the authors to discuss some studies on the effects of different substances, particularly statins as antioxidants and/or drugs that might interfere with the substances involved in the process of oxidation/antioxidation  (refences might be: Daliri M, et al.Effect of Statins on Superoxide Dismutase Level: A Systematic Review. Curr Med Chem. 2023 Aug 31. doi: 10.2174/0929867331666230831145809. AND Mansouri A, et al.Antioxidant Effects of Statins by Modulating Nrf2 and Nrf2/HO-1 Signaling in Different Diseases. J Clin Med. 2022 Feb 27;11(5):1313.). When discussing ROS, particularly in the context of prevention of cardiac fibrosis, I would suggest to the authors to discuss the antioxidant role of resveratrol (the reference could be: Zivarpour P, et al.Resveratrol and Cardiac Fibrosis Prevention and Treatment.Curr Pharm Biotechnol. 2022;23(2):190-200.).

See above

Author Response

Dear Reviewer,

Thank you very much for your thoughtful and constructive feedback. We are pleased that you found our review well-written, and we greatly appreciate your comments and suggestions.

We acknowledge your point regarding the number of self-citations and we reviewed and reduced them to ensure the inclusion of a broader and more balanced range of references.
Furthermore, we appreciate your recommendation to include studies on the effects of statins as antioxidants and their potential to interfere with oxidative processes. The suggested references (Daliri M, et al., 2023; Mansouri A, et al., 2022) are highly relevant, and we incorporated a discussion of these findings to enrich our analysis on the topic.
Regarding the role of reactive oxygen species (ROS), we also included a discussion of the antioxidant effects of resveratrol, as per your suggestion (Zivarpour P, et al., 2022). This will provide a more comprehensive view of the therapeutic potential of antioxidants in cardiac patologies prevention and treatment.
Please find the newly introduced paragraphs starting from line 94-132.
Thank you once again for your valuable input. We are confident that these changes will enhance the quality and depth of the manuscript.

Best regards on behalf of all authors, 
Alina Nicolae

Reviewer 2 Report

Redox Homeostasis and Molecular Biomarkers in Precision Therapy for Cardiovascular Diseases looks at the prospects for precision therapies for oxidant and oxidative stress conditions.  One immediate issue is the lack of mention of peroxiredoxins as a major antioxidant component.  This begins with the discussion centered around Fig. 1 where the intrinsic antioxidant system is divided as SOD, GPX, TRX, as opposed to either SOD, GPX, PRDX or SOD, GSH, TRX and omits catalase.  This carries over into the rest of the review.  The review, while well intentioned and making some valid points, requires some modifications to make it complete.

Starting on line 43, the introduction became a little confusing.  Please mention that ROS have effects in many systems, one of which is the nervous system, so that the reader can anticipate that more examples are to follow.

On line 155, is the activation of MMPs by oxidants direct or indirect?

On line 188, 8-iso----.

On line 193-195, mention that this non-enzymatic reaction is faster than SOD with superoxide.

On line 230, which NOXs are involved?

On line 244, Koppenol has consistently shown that the Haber-Weiss cycle is too slow to be considered a valid mechanism of oxidant generation.

Koppenol WH. The Haber-Weiss cycle--70 years later. Redox Rep. 2001;6(4):229-34. doi: 10.1179/135100001101536373. PMID: 11642713.

On line 267, GPX1 operates by a ping-pong mechanism, whereby the reaction with hydrogen peroxide is diffusion limited and the Km is contingent on the GSH levels; there is no fixed affinity that can bee compared to catalase, only rate constants can be compared.

 On line 357, define antioxidant deficiency as chemical, enzymatic or both.

On line 417 (N)itric

Author Response

Thank you very much for your detailed and insightful comments. We appreciate your thorough review and your suggestions to improve the quality and completeness of our manuscript.

Peroxiredoxins (PRDX) and Catalase Inclusion: We acknowledge the omission of peroxiredoxins and catalase as significant components of the intrinsic antioxidant system. We revised the discussion around Fig. 1 to reflect these important antioxidants and adjust the categorization accordingly. This modification will also be carried over throughout the manuscript to ensure a comprehensive representation of the antioxidant systems.

Clarifying the Role of ROS in Multiple Systems: Regarding the introduction starting on line 43, we appreciate your suggestion to clarify that ROS affect various systems, including the nervous system. We revised the text to include this broader perspective, allowing the reader to anticipate examples from multiple biological systems.

Clarification on the Activation of MMPs (Line 155): The activation of MMPs by oxidants can be both direct and indirect. We provided more details on this distinction and included relevant citations to clarify the underlying mechanisms.

Completion of Line 188 ("8-iso----"): Thank you for pointing out this sentence. We corrected the sentence to ensure clarity and completeness.

Non-Enzymatic Reaction Faster than SOD (Lines 193-195): We revised this section to explicitly mention that the non-enzymatic reaction is indeed faster than SOD with superoxide, providing more context and clarity to the discussion.

Identification of NOXs Involved (Line 230): We specified which NOX isoforms are involved in the particular processes mentioned, adding relevant details to avoid ambiguity.

Discussion of the Haber-Weiss Cycle (Line 244): We incorporated your recommendation regarding the Haber-Weiss cycle and deleted the paragraph regarding this confusing matter. This will add clarity and accuracy to the discussion.

Mechanism of GPX1 (Line 267): We appreciate your suggestion to elaborate on the ping-pong mechanism of GPX1. We revised the text to clarify that the reaction with hydrogen peroxide is diffusion-limited, and the Km is contingent on GSH levels.

Defining Antioxidant Deficiency (Line 357): We revised this section to define antioxidant deficiency more clearly, specifying whether it is chemical, enzymatic, or both, as per your suggestion.

Clarification on (N)itric (Line 417): We made the necessary correction on line 417 to properly address the term "(N)itric" and ensure clarity in the context provided.

Thank you again for your valuable feedback. We are confident that these revisions will significantly improve the manuscript and provide a more complete and accurate discussion on redox homeostasis and molecular biomarkers in precision therapy for cardiovascular diseases.

Reviewer 3 Report

Precision medicine has substantially improved the management of human diseases. Oxidative stress is implicated in the development of various human diseases such as cancer, diabetes, and cardiovascular diseases (CVD). Precision medicine approaches aim to identify specific oxidative stress biomarkers and genetic predispositions that contribute to these oxidative mechanisms. In this review article the authors compiled the various sources of reactive oxidant species (ROS) in the development of oxidative stress in CVD. This review helps the researchers/scientists to develop novel small molecule antioxidants as a “Therapy to Precision Medicine”.

Recent studies use human-derived organoids to advance human disease research, with a focus on promoting drug discovery and precision medicine. Human-derived organoids are useful to study the redox homeostasis and molecular biomarkers in precision therapy for CVD. Authors can comment about the role of organoids in the development of antioxidant therapies in precision medicine.

This review article is a valuable contribution in the field of cardiovascular diseases.

This manuscript/review has been well written and structured. It is easy to read and understand the concepts.

In Figure 3, complex II catalyzes the conversion of FADH2 to FAD. It is incorrect. Complex II catalyzes the oxidation of succinate to fumarate as shown in the literature (Erich Gnaiger, J. Biol. Chem. (2024) 300(1) 105470). Hence, the authors must correct it in the figure.

Author Response

Thank you for your thoughtful and encouraging comments on our manuscript. We greatly appreciate your positive feedback regarding the structure and clarity of the review, as well as your recognition of its contribution to the field of cardiovascular diseases.

We also appreciate your valuable suggestion to discuss the role of human-derived organoids in the development of antioxidant therapies in precision medicine. Organoids are indeed an emerging and powerful tool in understanding disease mechanisms, including redox homeostasis, and they hold great potential in advancing personalized therapies. We revised the manuscript and also we included a detailed discussion on how organoids can be used to study oxidative stress and their role in drug discovery for cardiovascular diseases. This addition will further enhance the relevance of our review in the context of precision medicine.

We also correct the figure 3 (Complex II catalyzes the oxidation of succinate to fumarate).

Thank you once again for your insightful feedback. We are confident that these changes improved the overall quality of the manuscript, and we look forward to submitting the revised version.

Round 2

Reviewer 1 Report

The authors have made all the changes suggested by the reviewers and therefore I have no further comments/suggestions.

See above.

Reviewer 2 Report

This submission is the revision to a paper I reviewed earlier.  In general the authors have revised the paper based on my suggestions, adequately.  I am not happy with Fig. 1 but that can be overlooked.

The authors made an honest attempt to revise the paper and it is acceptable in present form, although I would like to see figure revised to drop GPX from the fig and just include GSH, so the general division of hydroperoxide reduction is shown as GSH dependent, TRX dependent and catalase dependent.